# The Effects of Different Antimicrobial Peptides (A-11 and AP19) on Isolated Bacteria from Fresh Boar Semen and Semen Quality during Storage at 18 °C

**DOI:** 10.3390/antibiotics13060489

**Published:** 2024-05-24

**Authors:** Krittika Keeratikunakorn, Panida Chanapiwat, Ratchaneewan Aunpad, Natharin Ngamwongsatit, Kampon Kaeoket

**Affiliations:** 1Department of Clinical Sciences and Public Health, Faculty of Veterinary Science, Mahidol University, 999 Phuttamonthon 4 Rd., Salaya, Phuttamonthon, Nakhon Pathom 73170, Thailand; krittika.ker@student.mahidol.edu (K.K.); panida.chn@mahidol.edu (P.C.); natharin.nga@mahidol.edu (N.N.); 2Graduate Program in Biomedical Sciences, Faculty of Allied Health Sciences, Thammasat University, Rangsit Campus, Klongluang, Pathum Thani 12120, Thailand; aunpad@gmail.com; 3Laboratory of Bacteria, Veterinary Diagnostic Center, Faculty of Veterinary Science, Mahidol University, 999 Phuttamonthon 4 Rd., Salaya, Phuttamonthon, Nakhon Pathom 73170, Thailand

**Keywords:** antimicrobial peptides, boar semen, semen quality

## Abstract

Antibiotic resistance (AMR) is a major public health concern. Antimicrobial peptides (AMPs) could be an alternative to conventional antibiotics. The purpose of this research was to investigate the antimicrobial ability of the synthetic AMPs (i.e., A-11 and AP19) on the most frequently isolated bacteria in boar semen and their effect on extended boar semen quality during storage. We tested the antimicrobial effect of A-11 and AP19 at different concentrations and compared them with gentamicin for inhibiting the growth of E. coli, Pseudomonas aeruginosa and Proteus mirabilis that were isolated from fresh boar semen. In order to evaluate the effect of AMP on semen qualities on days 0, 1, 3, and 5 after storage at 18 °C, seven fresh boar semen samples were collected, diluted with semen extender with antibiotic (i.e., gentamicin at 200 µg/mL, positive control) or without (negative control), and semen extender contained only A-11 or AP19 at different concentrations (i.e., 62.50, 31.25, and 15.625 µg/mL). The total bacterial count was also measured at 0, 24, 36, 48, and 72 h after storage. Comparable to gentamicin, both A-11 and AP19 inhibited the growth of E. coli, Pseudomonas aeruginosa, and Proteus mirabilis at 62.50, 31.25, and 15.625 µg/mL, respectively. Comparing the total bacterial count at 0, 24, 36, 48 and 72 h after storage, the lowest total bacterial concentration was found in the positive control group (*p* < 0.05), and an inferior total bacterial concentration was found in the treatment groups than in the negative control. On day 1, there is a lower percentage of all sperm parameters in the AP19 group at a concentration of 62.50 µg/mL compared with the other groups. On day 3, the highest percentage of all sperm parameters was found in the positive control and A-11 at a concentration of 31.25 µg/mL compared with the other groups. The AP19 group at 62.5 µg/mL constantly yielded inferior sperm parameters. On day 5, only A-11 at a concentration of 15.625 µg/mL showed a total motility higher than 70%, which is comparable to the positive control. A-11 and AP19 showed antimicrobial activity against *E. coli*, *Pseudomonas aeruginosa* and *Proteus mirabilis* isolated from boar semen. Considering their effect on semen quality during storage, these antimicrobial peptides are an alternative to conventional antibiotics used in boar semen extenders. Nevertheless, the utilization of these particular antimicrobial peptides relied on the concentration and duration of storage.

## 1. Introduction

Artificial insemination (AI) has been used as assisted reproductive technology in the pig industry for many years [1]. Liquid boar semen preservation is commonly used for AI, and the semen extender must be provided for preserving semen [2,3]. The purpose of the semen extender is to protect sperm from cold shock, maintain pH and osmotic pressure, and inhibit bacterial growth, with the goal of preserving the longevity and quality of sperm [3]. Both Gram-positive and Gram-negative bacteria, such as *Streptococcus* spp., *Staphylococcus* spp., *E. coli*, *Klebsiella* spp., *Aeromonas* spp., *Pseudomonas* spp., *Proteus* spp., and *Providencia* spp. have been frequently found in fresh boar semen [4,5,6,7]. These abundances of bacteria are resident in boar’s skin, hair, and preputial diverticulum and contaminated into fresh boar semen during semen collection [8]. Bacteria contamination has several adverse impacts on the performance and quality of sperm as well as sow reproductive health [8]. In practice, many antibiotics are mixed into the semen extender to inhibit bacterial growth and limit the deleterious effect of this contamination [3,9,10]. Gentamicin, neomycin, streptomycin, and other antibiotics are commonly supplemented in boar semen extenders [3,11,12]. Further, more than one antibiotic is mixed with the boar semen extender, for example, gentamicin and polymyxin B or gentamicin and florfenicol, in order to inhibit both Gram-positive and Gram-negative bacteria [3,13]. Recently, it has been reported that the bacteria isolated from boar semen carried antibiotic resistance genes such as *mcr-3* and *int1* [7,14]. In addition, most bacteria from boar semen are prone to resistance to gentamicin and penicillin [15]. Antibiotic resistance is a worldwide problem owing to the overuse of unnecessary antibiotics in animals and humans, as well as the slow development of novel antibiotic discoveries [14].

Many studies have been performed on substitute strategies that can lower the usage of antibiotics in pig farms, including reducing or replacement the antibiotic supplementation in boar semen extenders. Antimicrobial peptides (AMPs) have been determined to be an alternative antimicrobial agent of interest, in which it showed compromised results for inhibiting *Escherichia coli* isolated from boar semen that carry antibiotic resistance genes [16]. To date, it has been documented that altogether 3257 AMPs were added to the Antimicrobial Peptide Database (APD) [17]. Most AMPs have been discovered and identified as antimicrobial agents, and can be applied for the treatment of antibiotic-resistant bacteria [16,18]. These include proline-rich antimicrobial peptides (PrAMPs), tryptophan- and arginine-rich antimicrobial peptides, histidine-rich antimicrobial peptides, and glycine-rich antimicrobial peptides [19,20]. The differences in the charge between the membranes of animals and bacteria can enable AMPs to become active through direct and rapid binding to the outer bacterial cell wall, such as lipopolysaccharide (LPS) in Gram-negative bacteria or teichoic acid in Gram-positive bacteria [14,21,22,23]. Additionally, the outermost surface of bacterial cells contains lipopolysaccharides, or teichoic acid [21,24,25]. The positive charge of AMPs strongly interacts with the negative charge there, but it has a weak interaction with the positively charged animal membrane [20,21,22,23,24]. More significantly, the key characteristic of AMPs is their capacity to kill bacteria without damaging the host cell [26]. Therefore, AMPs is an interesting choice to reduce or replace antibiotic usage in boar semen extender. A-11 and AP19 are two novel AMPs, when used in high concentrations, are not damaging animal cells and inhibiting the growth of both Gram-positive and Gram-negative bacteria, including *Salmonella enterica* serovar Typhimurium and *Acinetobacter baumannii* [27,28]. However, the application of these two peptides on the inhibition of bacteria isolated from boar semen has not been reported.

The purpose of this study was to determine the antimicrobial ability of A-11 and AP19 whether to inhibit the growth of most frequently found bacteria (i.e., *E. coli*, *Pseudomonas aeruginosa* and *Proteus mirabilis*) in boar semen and, subsequently, their effect on boar semen quality while being used as a replacement of antibiotics in boar semen extender.

## 2. Results

### 2.1. Bacterial Survival Assay

The growth curves of *E. coli*, *Pseudomonas aeruginosa*, and *Proteus mirabilis* are shown in Figure 1, Figure 2 and Figure 3. Similar to gentamicin, A-11 and AP19 showed their ability to inhibit the growth of *E. coli* (Figure 1). After 12 h of growth, the OD_600_ values of *E. coli* in the gentamicin group increased, while A-11 (Figure 1A) and AP19 (Figure 1B) plateaued. In addition, the OD_600_ value of *E. coli* at 14 h in the A-11 at 15.625 µg/mL was slightly increased (Figure 1A). *Pseudomonas aeruginosa* was inhibited by gentamicin and AP19 (Figure 2B) throughout the investigation period. However, A-11 at 15.625 µg/mL was able to inhibit *Pseudomonas aeruginosa* only for 10 h (Figure 2A). Similar to *E. coli* and *Pseudomonas aeruginosa*, *Proteus mirabilis* was inhibited by gentamicin and AP19 throughout the investigation period (Figure 3B). However, A-11 at 15.625 µg/mL was able to inhibit *Proteus mirabilis* only for 16 h (Figure 3A). The OD_600_ value of *Proteus mirabilis* at 21 h in the A-11 at 31.25 µg/mL was also slightly increased (Figure 3A).

### 2.2. Sperm Quality Parameters Analysis

The sperm quality of fresh boar semen samples is presented in Table 1. On day 1 (Table 2), the sperm quality parameters remained normal, and there was no significant difference in all sperm parameters among the 8 groups. However, there were inferior values for all sperm parameters, particularly viability and MMP, in AP19 at a concentration of 62.50 µg/mL than in other groups. On day 3 (Table 3), no significant difference was observed when compared all sperm parameters in AP19 at a concentration of 31.25 µg/mL with the positive control group. In addition, there was a significantly lower percentage for all sperm quality parameters of AP19 at a concentration of 62.50 µg/mL (*p* < 0.05). The A-11 and AP19 at the same concentration of 15.625 µg/mL yielded an acceptable percentage of more than 70 in terms of total motility, viability and intact acrosome. On day 5 (Table 4), only total motility, viability and intact acrosome parameters in the positive control, negative control and AP19 at a concentration of 31.25 µg/mL remained higher than 70%.

### 2.3. *Total Bacterial Concentration*

The mean total bacterial concentration of fresh boar semen was log2.36 ± 0.5 CFU/mL (ranged log1.74 to log 3.04 CFU/mL) (Table 1). The total bacterial concentration of diluted semen at 0, 24, 36, 48, and 72 h after storage at 18 °C is shown in Table 5. The total bacterial concentration increased as the incubation period was prolonged. At 0 h after incubation, the highest total bacterial concentration was found in the negative control group (log1.32 CFU/mL, BTS without antibiotic) when compared with other groups. Comparing among the AMP groups, AP19 at a concentration of 31.25 µg/mL showed a higher bacterial concentration than the other groups (i.e., log0.78 CFU/mL). At 24 h after storage, the total bacterial concentration of the positive control group (BTS with gentamicin) was still absent, while the concentration in other groups continued increasing. However, the negative control group (BTS without antibiotic) had the highest total bacterial concentration (log2.38 CFU/mL), compared with the AMPs and positive control groups. At 36 h after storage, the highest total bacteria concentration was found in the negative control group (log3.11 CFU/mL, BTS without antibiotic) when compared with other groups. However, there was no significant difference in the bacterial concentration among the AMP groups which varied from log2.47 to 2.78 CFU/mL. At 48 h after storage, the lowest bacterial concentration among the AMP groups was found in AP19 at a concentration of 15.625 µg/mL (log2.86 CFU/mL), while the bacterial concentrations in the negative and positive control were log3.71 CFU/mL and absent, respectively. At 72 h after storage, the bacterial concentration among the AMP groups varied from log4.54 to 4.96 CFU/mL, while the highest bacterial concentration was still found in the negative control group (i.e., log5.31 CFU/mL).

## 3. Discussion

The concentration of AMPs (A-11 and AP19) for this study came from the MIC value (62.50–15.625 µg/mL) for inhibiting Gram-negative bacteria in the previous studies [27,28] and was further approved for inhibiting the most Gram-negative bacteria observed in fresh semen by comparing with 200 µg/mL of gentamicin [7], which is the common antibiotic mixed in boar semen extenders [13]. The results of the bacterial survival assay clearly showed the inhibitory effect of AMPs on bacterial growth in each stage of bacterial growth curve. The current findings regarding the total bacterial count clearly demonstrate that the A-11 and AP19 peptides have the ability to inhibit bacterial growth for a minimum of 36 h when stored at 18 °C. During this period, the total bacterial concentration in all treatment groups remained below log2.80 CFU/mL (ranging from 2.47 to 2.78), in contrast to the log3.11 CFU/mL observed in the negative control. Ciornei et al. [29] determined that the normal range for the overall bacterial concentration in fresh boar semen is between 22.40 and 188.20 × 10^3^ CFU/mL (equivalent to log4.35–5.27) for optimal reproductive outcomes in pig farming. According to reports, there was a 6.4% reduction in sperm viability for every log_10_ increase in total bacterial concentration [30]. Furthermore, if boar semen was found to have a contamination level of *E. coli* exceeding 3.5 × 10^3^ CFU/mL (log3.54), it led to a reduction in litter size and consequently had a negative impact on reproductive performance in pig farms [31]. The primary cause of sow endometritis or post-mating vaginal discharge is typically the presence of *E. coli* contamination in boar semen prior to artificial insemination [4,32,33]. This condition, known as acute endometritis, has the potential to progress into chronic endometritis, which can then have a negative impact on the reproductive performance of pigs [31]. In addition to *E. coli*, recent reports have indicated that *Pseudomonas aeruginosa* and *Proteus mirabilis* are the most common bacteria found in fresh boar semen [7]. In this study, it is noteworthy that A-11 and AP19 effectively inhibited the growth of contaminating bacteria in semen samples, regardless of the concentration of antimicrobial peptides used. Importantly, this inhibition did not have any adverse effects on the quality of the semen.

This study used a short-term semen extender (BTS), which has the ability to preserve semen quality of less than or equal to three days after dilution [34]. For the reasons mentioned, this study observed the sperm quality at days 0, 1, 3, and 5 during storage to ensure that the BTS still maintained sperm quality as claim by the manufacturer. While the bacteria growth during storage was rapid growth and significant growth after 72 h of storage, as a result, the total bacteria concentration was measured at 0, 24, 36, 48, and 72 h of storage [4,8,35]. In practice for the pig farms, it is also worth noting that they usually used extended boar semen within 24 h after storage. Consequently, the present experimental design was correspondent to those clinical practices. When examining semen qualities, specifically total motility and progressive motility, after being stored at 18 °C from days 0 to 5 in all groups, it was found that the total bacterial count increased over time. However, the negative impact on semen qualities was only observed when the semen extender was supplemented with a high concentration of A-11 and AP19 (62.50 µg/mL). The observed effect was evident on day 3 for AP19 and on day 5 for A-11. The semen extender used in this study is BTS based and specifically designed for short-term preservation of boar semen, with a recommended storage period of 3 days. After evaluating the semen quality on day 3 following storage, it was found that only A-11 at a concentration of 31.25 µg/mL produced semen quality similar to that of the positive control group. The present results of A-11 clearly showed that there is no sign of toxicity to sperm cells for all concentrations. This is in agreement with the hemolytic activity examination of A-11, which discovered that A-11 did not cause damage to red blood cells at concentrations between 0.98 and 250 µg/mL [27]. The underlying mechanism might be that in the outer membrane of animal cell (i.e., sperm cell) constituent of neutral components, subsequently the positively charge of AMP were not interaction with this cell [14,22].

Collectively, the antimicrobial peptides employed in this investigation demonstrate the capacity to impede bacterial proliferation within the initial 36 h period and sustain the quality of boar semen for a duration of 3 days. This phenomenon can be attributed to the interaction between positively charged antimicrobial peptides and the negatively charged teichoic acid or lipopolysaccharides present on the outermost membrane of bacterial cells [21,22,24]. The negative charge of the animal cell membrane is situated internally and in close proximity to the cytoplasm and in the outer membrane of were expressed neutral components. Consequently, the positively charged antimicrobial peptides do not interact with this cell [14,22,27]. Prior research has shown that the rupture of the *E. coli* membrane is triggered by the difference in the charge between animal and bacterial cell membranes. This allows active AMPs to exclusively bind to the bacterial membrane, leading to membrane dysfunction. This dysfunction is caused by the induction of membrane curvature, the formation of membrane pores, and ultimately the lysis of the bacterial cell [14,17,22,23,24]. At the optimal concentrations, AMPs caused damage to bacterial cell membranes. However, at lower concentrations, they moved into the cytoplasm and engaged in electrostatic interactions with bacterial DNA or ribosomes [36,37,38]. As stated by Schulze et al. [39], a high concentration of AMPs can have a detrimental effect on spermatozoa, which aligns with the findings of this study. The two AMPs examined in this study exhibited contrasting outcomes in terms of their ability to inhibit bacterial growth and preserve semen quality. These differences can potentially be attributed to their varying hydrophobicity levels (A-11 = 44% and AP19 = 47%), which may also contribute to their toxicity towards sperm cells. Hydrophobicity plays a role in the effectiveness and specificity of AMPs in interacting with the target cell. This character facilitates the incorporation of water-soluble AMPs into the lipid bilayer of the membrane. The activity and selectivity of a substance are determined by its hydrophobicity. A high level of hydrophobicity can be harmful to the animal cell membrane and reduce antimicrobial activity [40,41]. In order to prevent the use of excessively high concentrations of AMPs, it has been shown that combining antimicrobial peptides with antibiotics can reduce the negative effects on boar sperm. For instance, in liquid-stored boar semen, a combination of 0.16 g/L epsilon-polylysine (ε-PL) and 0.125 g/L gentamicin resulted in similar sperm quality compared to using 0.25 g/L gentamicin alone. Studies have reported using a combination of two distinct AMPs or a combination of an AMP and antibiotics to address the issue of multidrug-resistant bacteria [42]. However, it is crucial to note that one of the key features of the AMPs utilized in boar semen extenders is its ability to prevent bacterial growth without damaging spermatozoa [39,43,44]. Additional research is required to examine the impact of A-11 and AP19 on farm fertility, specifically in relation to post-mating vaginal discharge, pregnancy rate, farrowing rate, and litter size, before introducing these peptides into the pig industry.

## 4. Materials and Methods

### 4.1. Synthesis of Peptides and Their Physical-Chemical Analysis

The AMPs in this study were synthesized, determined for physicochemical properties (Table 6) and validated for inhibiting *Pseudomonas aeruginosa* isolated from human clinical cases by Klubthawee et al. [45]. In brief, after being created utilizing solid-phase methods and 9-fuorenylmethoxycarbonyl (Fmoc) chemistry, the A-11 and AP19 peptides were synthesized by solid-phase techniques and purified as trifluoroacetate salts by HPLC (ChinaPeptides, Shanghai, China). 19F nuclear magnetic resonance (NMR) revealed that there was less than 1.7% (wt/wt) of residual TFA present. The TAMRA-labeled antimicrobial peptide was created via dehydration condensation, and TAMRA was bound to antimicrobial peptide via an amide bond at the N-terminus. Reversed-phase HPLC analysis revealed that more than 98% of the peptides were purified. Electrospray Ionization Mass Spectrometry (ESI-MS) was used to identify the peptides [45].

### 4.2. Bacterial Strains and Culture Conditions

The bacteria in the present study were obtained from our previous report by Keeratikunakorn et al. [7] in which three species of the most frequently found bacteria in fresh boar semen including *E*. *coli*, *Pseudomonas aeruginosa*, and *Proteus mirabilis* were isolated and kept in a culture collection at the Laboratory of Bacteria, Veterinary Diagnostic Center, Faculty of Veterinary Science, Mahidol University (Salaya, Phuttamonthon, Nakhon Pathom, Thailand). The bacteria *E*. *coli*, *Pseudomonas aeruginosa*, and *Proteus mirabilis* were all grown in a brain heart infusion (BHI, Difco, Reno, NV, USA) medium and incubated for 16–18 h at 37 °C. Pre-culture was performed by inoculating BHI broth with a single isolated colony and then shaking it at 200 rpm for 16–18 h at 37 °C. Before being used, a 1% concentration of the pre-culture was added to the BHI broth and kept to grow at 37 °C.

### 4.3. Bacterial Survival Assay

The bacteria were cultivated in BHI broth before being moved to a normal saline solution (0.85% NaCl) to achieve the 0.5 McFarland standard (10^8^ CFU/mL). A 500 µL bacterial suspension diluted to 10^6^ CFU/mL in Mueller–Hinton broth (Difco^TM^, Reno, NV, USA), was used in each well of the triplicate experiments, which used 48-well plates. This was mixed with 500 μL of appropriate antimicrobial peptide dilutions at the doses of 62.50, 31.25. 15.625, 0 µg/mL (growth control). A positive control, gentamicin 200 µg/mL was used. The OD_600_ values were measured every hour for a 24 h period at 37 °C using a microplate spectrophotometer (BMG LABTECH, SPECTROstar Nano, Ortenberg, Germany), and a growth curve was created [17].

### 4.4. Boar Semen Collection and Preparation

A semen sample was collected from each of the seven mature Duroc boars, theirs ages ranged from 1.5 to 3 years. Boar semen was collected using the gloved-hand technique [46]. Only the sperm-rich fractions of the semen were collected after it was filtered via gauze. The sperm motility, concentration, percentage of viability, intact acrosome, sperm with high mitochondrial membrane potential, osmolality and total bacterial concentration of the fresh semen were measured after collection [42]. Only semen ejaculates with a progressive motility of more than 70% and a concentration more than 100 × 10^6^ spermatozoa/mL were included in the experiment [47].

As shown in Table 7, the fresh boar semen was divided into 8 groups and diluted with different semen extenders as follows: Beltsville Thawing Solution with 200 µg/mL of gentamicin (BTS; Minitube, Tiefenbach, Germany), BTS without antibiotic (Minitube, Tiefenbach, Germany), and BTS without antibiotic plus various concentrations of A-11 and AP19. The sperm concentration was adjusted to 3.0 × 10^9^ sperm/100 mL. The diluted semen samples were stored in a digitally controlled refrigerator at 18 °C until evaluation. The total bacterial concentration was assessed at 0, 24, 36, 48, and 72 h after storage. The quality of sperm was evaluated on days 1, 3, and 5 after storage.

### 4.5. Sperm Parameters Analysis

#### 4.5.1. Total Motility and Progressive Motility

The sperm motility was analyzed by computer-assisted sperm motility analysis (CASA) (AndroVision^®^, Minitube, Tiefenbach, Germany). In summary, 3 µL of expanded semen was inserted into the counting chamber (Leja^®^, IMV Technologies, L’Aigle, Basse-Normandie, France) in which the temperature of glass slide and stage were set at 37 °C. The data were then recorded right away using the CASA. Each sample has five fields that are evaluated, and each analysis counts at least 600 cells. The percentage of motile sperm, progressive motile sperm, and motility patterns, including curvilinear velocity (VCL, µm/s), average pathway velocity (VAP, mm/s), straight-line velocity (VSL, mm/s), beat cross frequency straightness (BCF, Hz), amplitude of lateral head displacement (ALH, mm), straightness (STR; VSL/VAP, %), and linearity (LIN; VSL/VCL, %) were expressed in the analysis results [46].

#### 4.5.2. Sperm Acrosomal Integrity

To assess acrosomal integrity, fluorescein isothiocyanate-labeled peanut (*Arachis hypogaea*) agglutinin (FITC-PNA) with staining was employed. A total of 10 µL of the semen sample was incubated with 10 µL of EthD-1 at 37 °C for 15 min A glass slide was covered with the mixture in 5 µL, air-dried, and fixed in 95% ethanol for 30 sec. Spread all through the slide, 50 µL of diluted FITC-PNA (diluted with PBS 1:10 *v*/*v*) was incubated at 4 °C in a moist chamber for 30 min. Thereafter, the slide was washed with cold PBS and given to air dry. Under a fluorescence microscope, 200 sperm were examined and separated into two groups: those with intact and damaged acrosomes [34,46,47].

#### 4.5.3. Sperm Viability

SYBR-14 (L7011; Live/Dead™ Sperm viability kit, Invitrogen, Waltham, MA, USA) and Ethidiumhomodimer-1 (EthD-1, E1169, Invitrogen, Waltham, MA, USA) staining were performed to assess the sperm viability. The mixture contained 10 µL of semen sample, 2.7 µL of SYBR-14 (0.54 μM in DMSO), and 10 µL of EthD-1 (1.17 μM in PBS). The mixture was incubated at 37 °C for 20 min. After incubation, 5 µL of the processed sample was pipetted to the glass slide, and the coverslip was placed over it. A fluorescent microscope with a 1000× magnification was used to examine 200 sperm, which were then separated into live and dead sperm [46].

#### 4.5.4. Sperm with High Mitochondrial Membrane Potential (MMP)

The membrane potential of the mitochondria was assessed by a staining approach with fluorochrome5,5′,6,6′-tetrachloro-1,1′,3,3′-tetraethylbenzimidazolyl-carbocyanine iodide (JC-1; T3168, Invitrogen, Waltham, MA, USA). The mixture contained 50 µL of diluted semen samples, 3 µL of a 2.4 mM propidium iodide (PI) solution, and 3 µL of a 1.53 mM JC-1 solution in DMSO and was then incubated at 37 °C for 10 min. Two hundred live sperm (PI-negative) were examined using a 400× magnification fluorescent microscope and identified as having high mitochondrial membrane potential (yellow-orange fluorescence) and low mitochondrial membrane potential (green fluorescence) [46].

### 4.6. Total Bacterial Concentration

The spread plate technique was used to determine the total bacterial content following to the incubation of the boar semen samples at 18 °C. Using a ten-fold dilution method, semen samples were diluted with 0.85% NaCl. Each dilution of a semen sample was spared 100 µL, and these were then cultured at 37 °C on plate count agar (PCA, DifcoTM, Reno, NV, USA). After incubation, the colonies were enumerated and converted to CFU/mL at 48 h [48].

### 4.7. Statistical Analysis

The statistical analysis was performed by using PASW Statistics for Windows, version 18.0 (SPSS Inc., Chicago, IL, USA). The normal distribution test of the data was evaluated using the Shapiro–Wilk test. The total bacterial concentration was presented as the mean ± SD and semen parameters data were presented as the mean ± SEM. The bacterial concentration and sperm parameters data analysis were performed by using one-way analysis of variance (ANOVA) and compared means by using Duncan’s test. *p* < 0.05 was considered statistically significant.

## 5. Conclusions

The antimicrobial peptides A-11 and AP19 demonstrated the capacity to inhibit the growth of *E. coli*, *Pseudomonas aeruginosa*, and *Proteus mirabilis* that were obtained from fresh boar semen and extended boar semen stored at 18 °C. The potential of these peptides as an alternative to antibiotics in boar semen extenders is being remarked. Nevertheless, the utilization of these specific antimicrobial peptides relied on the concentration and duration of storage.

## Figures and Tables

**Figure 1 antibiotics-13-00489-f001:**
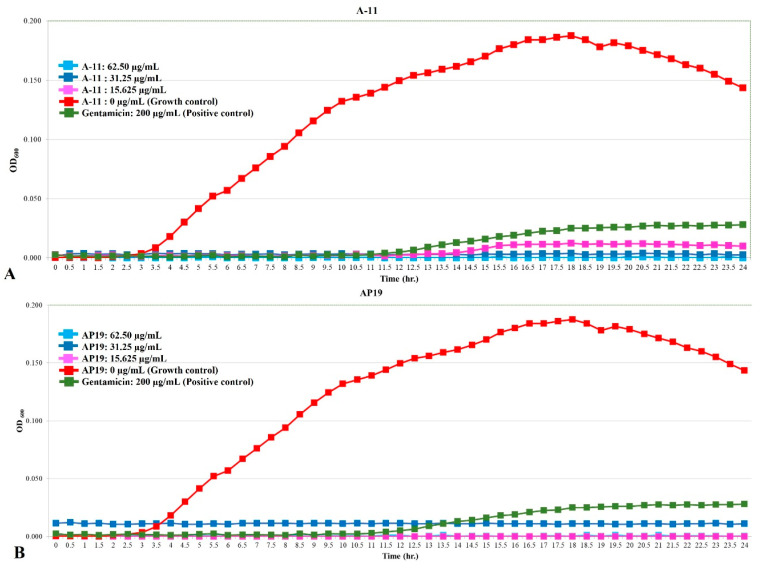
The growth curve of E. coli incubated with different concentrations of A-11 (**A**) and AP19 (**B**).

**Figure 2 antibiotics-13-00489-f002:**
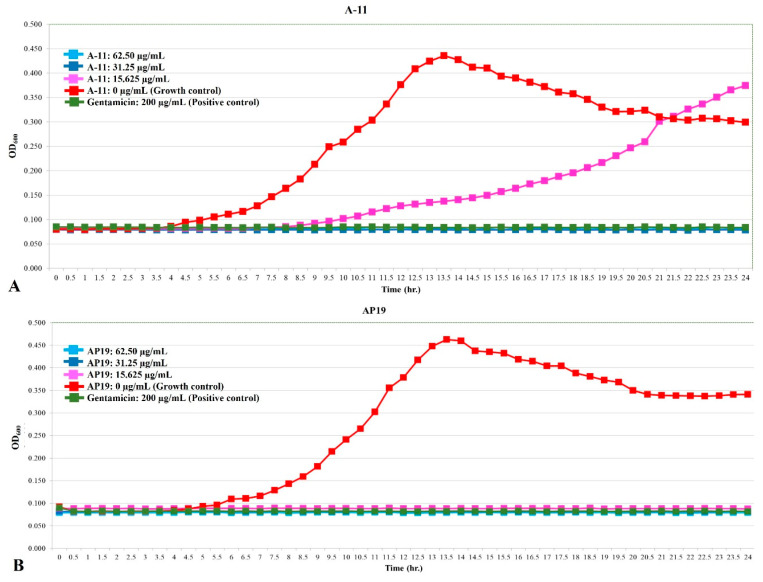
The growth curve of Pseudomonas aeruginosa incubated with different concentrations of A-11 (**A**) and AP19 (**B**).

**Figure 3 antibiotics-13-00489-f003:**
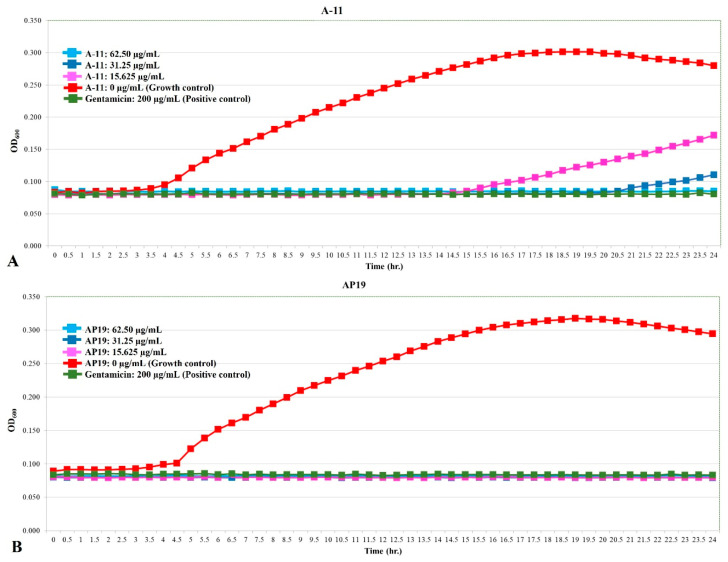
The growth curve of *Proteus mirabilis* incubated with different concentrations of A-11 (**A**) and AP19 (**B**).

**Table 1 antibiotics-13-00489-t001:** Descriptive statistics for sperm parameters measurements of fresh boar semen (n = 7).

Parameters	Mean ± S.D.	Range
Concentration (×10^6^ sperm/mL)	381.9 ± 72.4	292–495
Osmolality (mOsm/kg)	308.80 ± 3.69	305–316
Total motility (%)	90.3 ± 2.22	87.4–94.1
Progressive motility (%)	84.5 ± 3.2	79.7–88.3
Sperm viability (%)	88.2 ± 2.7	85–91
Intact acrosome (%)	88.2 ± 4.5	84–95
MMP (%)	86.1 ± 1.7	83–88
Total bacterial count (log_10_; CFU/mL)	2.36 ± 0.51	1.74–3.04

MMP: sperm with high mitochondrial membrane potential.

**Table 2 antibiotics-13-00489-t002:** Means ± SEM of semen quality parameters on day 1 after storage at 18 °C (n = 7).

Sperm Parameters	Groups							
	BTS	BTS + ABO	A-11	A-11	A-11	AP19	AP19	AP19
			62.50 µg/mL	31.25 µg/mL	15.625 µg/mL	62.5 µg/mL	31.25 µg/mL	15.625 µg/mL
MOT (%)	86.1 ± 1.9	87.6 ± 2.3	87.0 ± 1.9	87.0 ± 1.1	85.3 ± 1.7	82.1 ± 3.1	85.1 ± 3.0	87.2 ± 1.7
PMOT (%)	79.7 ± 2.3	81.1 ± 3.1	80.0 ± 2.2	80.2 ± 1.8	77.8 ± 1.9	72.3 ± 4.2	78.5 ± 4.2	80.7 ± 2.0
VCL (µm/s)	98.3 ± 16.5	89.7 ± 4.3	90.8 ± 10.5	88.7 ± 4.9	84.2 ± 3.8	76.0 ± 5.1	82.7 ± 6.9	89.6 ± 3.8
VSL (µm/s)	25.3 ± 2.6	26.1 ± 2.0	25.9 ± 2.1	26.6 ± 2.9	23.5 ± 2.6	22.2 ± 3.7	26.1 ± 3.3	26.9 ± 1.8
VAP (µm/s)	37.6 ± 1.2	36.4 ± 1.9	35.9 ± 2.2	36.1 ± 3.2	32.2 ± 2.6	30.3 ± 2.3	34.4 ± 3.7	36.9 ± 1.9
ALH (µm)	0.91 ± 0.04	0.94 ± 0.06	0.92 ± 0.05	0.90 ± 0.03	0.87 ± 0.03	0.80 ± 0.05	0.85 ± 0.07	0.92 ± 0.04
BCF (Hz)	16.4 ± 1.2	16.9 ± 1.3	17.6 ± 0.9	17.5 ± 1.2	16.7 ± 1.1	15.9 ± 1.3	16.8 ± 1.3	17.4 ± 1.2
STR (%)	70.1 ± 3.8	71.1 ± 3.3	71.3 ± 2.7	72.7 ± 3.2	71.7 ± 2.9	72.4 ± 3.2	69.9 ± 3.9	72.5 ± 2.1
LIN (%)	29.4 ± 2.8	29.4 ± 2.5	28.6 ± 1.8	29.6 ± 2.3	27.6 ± 2.5	28.5 ± 2.7	32.8 ± 3.8	29.8 ± 1.6
Viability (%)	84.9 ± 1.5 ^a,b^	86.4 ± 1.0 ^b^	81.5 ± 0.8 ^a,b^	85.6 ± 1.8 ^a,b^	83.5 ± 0.7 ^a,b^	80.7 ± 2.0 ^a^	82.6 ± 2.2 ^a,b^	83.2 ± 1.2 ^a,b^
Intact acrosome (%)	81.3 ± 1.6	81.9 ± 1.6	80.1 ± 1.8	80.2 ± 1.8	78.1 ± 1.8	77.0 ± 2.2	79.1 ± 2.0	79.9 ± 1.1
MMP (%)	78.2 ± 1.0 ^b^	78.6 ± 5.2.1 ^b^	78.3 ± 1.7 ^b^	80.0 ± 1.3 ^b^	76.6 ± 1.6 ^a,b^	72.4 ± 2.3 ^a^	76.9 ± 2.3 ^a,b^	77.9 ± 1.1 ^a,b^

Values in each row marked with different superscript letters differ significantly (*p*-value < 0.05). ABO: antibiotic (gentamicin 200 µg/mL); BTS: Beltsville Thawing Solution; MOT: total motility; PMOT: progressive motility; VCL: curvilinear velocity; VSL: velocity straight line; VAP: average pathway velocity; ALH: amplitude of lateral head displacement; BCF: beat cross frequency, straightness; STR: straightness; LIN: linearity; MMP: sperm with high mitochondrial membrane potential.

**Table 3 antibiotics-13-00489-t003:** Means ± SEM of semen quality parameters on day 3 after storage at 18 °C (n = 7).

Sperm Parameters	Groups							
	BTS	BTS + ABO	A-11	A-11	A-11	AP19	AP19	AP19
			62.50 µg/mL	31.25 µg/mL	15.625 µg/mL	62.5 µg/mL	31.25 µg/mL	15.625 µg/mL
MOT (%)	80.0 ± 2.2 ^b^	80.0 ± 2.2 ^b^	76.6 ± 4.0 ^b^	82.0 ± 2.0 ^b^	78.9 ± 4.0 ^b^	60.9 ± 8.2 ^a^	76.5 ± 4.0 ^b^	79.1 ± 3.0 ^b^
PMOT (%)	68.6 ± 2.9 ^b^	72.0 ± 3.7 ^b^	65.9 ± 5.5 ^b^	71.8 ± 2.9 ^b^	68.0 ± 4.8 ^b^	48.8 ± 9.4 ^a^	64.7 ± 4.7 ^b^	67.8 ± 4.3 ^b^
VCL (µm/s)	78.3 ± 3.1 ^a^	83.7 ± 3.9 ^a^	72.8 ± 6.7 ^a,b^	78.1 ± 5.3 ^a^	72.0 ± 7.2 ^a,b^	57.4 ± 10.8 ^b^	74.1 ± 8.3 ^a,b^	77.5 ± 5.5 ^a^
VSL (µm/s)	21.2 ± 1.6	21.8 ± 2.1	20.6 ± 2.3	21.4 ± 2.3	23.5 ± 2.6	15.3 ± 3.4	19.5 ± 3.5	20.6 ± 2.2
VAP (µm/s)	30.7 ± 2.1	30.7 ± 2.1	28.8 ± 2.9	30.1 ± 3.0	32.2 ± 2.6	21.5 ± 4.6	28.0 ± 4.4	29.4 ± 2.7
ALH (µm)	0.87 ± 0.03 ^a,b^	0.90 ± 0.04 ^a^	0.80 ± 0.07 ^a,b^	0.82 ± 0.04 ^a,b^	0.87 ± 0.03 ^a,b^	0.65 ± 0.09 ^b^	0.78 ± 0.06 ^a,b^	0.82 ± 0.05 ^a,b^
BCF (Hz)	13.7 ± 0.9 ^a,b^	15.2 ± 0.9 ^a^	13.2 ± 1.4 ^a,b^	15.8 ± 1.2 ^a^	16.7 ± 1.1 ^a^	10.9 ± 2.0 ^b^	14.4 ± 1.5 ^a,b^	14.3 ± 1.2 ^a,b^
STR (%)	69.0 ± 1.7	68.0 ± 1.8	71.6 ± 1.8	70.7 ± 1.4	71.7 ± 2.9	69.3 ± 2.2	68.4 ± 2.3	69.5 ± 1.8
LIN (%)	26.9 ± 1.1	25.7 ± 1.4	28.1 ± 1.3	27.0 ± 1.4	27.6 ± 2.5	25.7 ± 1.8	25.1 ± 1.8	26.3 ± 1.7
Viability (%)	82.4 ± 0.7 ^b^	82.4 ± 0.8 ^b^	76.5 ± 2.7 ^a,b^	81.0 ± 0.6 ^b^	78.6 ± 2.7 ^b^	69.8 ± 5.0 ^a^	76.7 ± 2.5 ^a,b^	79.4 ± 1.9 ^b^
Intact acrosome (%)	77.9 ± 1.7	78.0 ± 1.4	76.8 ± 1.4	77.0 ± 1.4	76.6 ± 1.1	73.6 ± 2.0	76.1 ± 1.3	75.4 ± 1.1
MMP (%)	69.4 ± 1.3 ^b^	70.5 ± 1.5 ^b^	66.2 ± 2.3 ^b^	70.3 ± 2.0 ^b^	68.3 ± 3.0 ^b^	53.5 ± 6.3 ^a^	67.4 ± 2.0 ^b^	68.3 ± 2.1 ^b^

Values in each row marked with different superscript letters differ significantly (*p*-value < 0.05). ABO: antibiotic (gentamicin 200 µg/mL); BTS: Beltsville Thawing Solution; MOT: total motility; PMOT: progressive motility; VCL: curvilinear velocity; VSL: velocity straight line; VAP: average pathway velocity; ALH: amplitude of lateral head displacement; BCF: beat cross frequency, straightness; STR: straightness; LIN: linearity; MMP: sperm with high mitochondrial membrane potential.

**Table 4 antibiotics-13-00489-t004:** Means ± SEM of semen quality parameters on day 5 after storage at 18 °C (n = 7).

Sperm Parameters	Groups							
	BTS	BTS + ABO	A-11	A-11	A-11	AP19	AP19	AP19
			62.50 µg/mL	31.25 µg/mL	15.625 µg/mL	62.5 µg/mL	31.25 µg/mL	15.625 µg/mL
MOT (%)	74.5 ± 4.6 ^b^	77.4 ± 5.9 ^b^	63.8 ± 7.5 ^a,b^	67.1 ± 7.8 ^a,b^	70.2 ± 5.9 ^a,b^	48 ± 9.3 ^a^	55.6 ± 8.0 ^a,b^	60.7 ± 8.4 ^a,b^
PMOT (%)	58.3 ± 7.0 ^a,b^	64.0 ± 7.1 ^a,b^	51.2 ± 7.5 ^a,b^	54.4 ± 9.2 ^a,b^	58.9 ± 7.7 ^a,b^	36.8 ± 8.0 ^a^	42.9 ± 8.1 ^a,b^	47.5 ± 9.0 ^a,b^
VCL (µm/s)	71.6 ± 9.4 ^a^	81.3 ± 13.0 ^a^	62.2 ± 9.5 ^a,b^	63.9 ± 12.0 ^a,b^	67.4 ± 10.3 ^a,b^	43.6 ± 8.4 ^b^	55.2 ± 9.8 ^a,b^	57.7 ± 9.9 ^a,b^
VSL (µm/s)	18.6 ± 3.8 ^a,b^	23.3 ± 5.4 ^a^	18.1 ± 3.6 ^a,b^	17.6 ± 4.5 ^a,b^	22.1 ± 4.3 ^a^	11.6 ± 2.7 ^b^	15.6 ± 3.6 ^a,b^	15.5 ± 3.7 ^a,b^
VAP (µm/s)	26.3 ± 4.8 ^a,b^	33.0 ± 6.6 ^a^	25.2 ± 4.6 ^a,b^	25.0 ± 5.7 ^a,b^	29.4 ± 5.4 ^a,b^	17.3 ± 3.8 ^b^	22.5 ± 4.9 ^a,b^	23.1 ± 5.0 ^a,b^
ALH (µm)	0.78 ± 0.09 ^a,b^	0.86 ± 0.11 ^a^	0.73 ± 0.09 ^a,b^	0.73 ± 0.11 ^a,b^	0.76 ± 0.11 ^a,b^	0.55 ± 0.09 ^b^	0.67 ± 0.10 ^a,b^	0.71 ± 0.10 ^a,b^
BCF (Hz)	11.9 ± 0.9 ^a,b^	14.0 ± 1.3 ^a^	9.5 ± 1.2 ^b,c^	10.0 ± 1.6 ^b,c^	12.7 ± 1.7 ^a,b^	6.8 ± 1.4 ^c^	8.3 ± 1.2 ^b,c^	9.3 ± 1.6 ^b,c^
STR (%)	69.6 ± 1.8	69.0 ± 3.2	71.0 ± 2.2	68.6 ± 2.2	73.5 ± 2.4	63.7 ± 3.0	68.0 ± 2.7	65.4 ± 3.2
LIN (%)	24.7 ± 1.9	27.1 ± 2.1	28.3 ± 1.7	25.6 ± 1.7	31.3 ± 3.4	24.1 ± 2.2	26.1 ± 2.2	24.8 ± 2.4
Viability (%)	77.6 ± 2.1 ^a,b^	77.8 ± 2.3 ^b^	68.0 ± 5.3 ^a,b^	75.0 ± 2.9 ^b^	73.9 ± 3.6 ^b^	58.1 ± 7.3 ^a^	65.0 ± 5.8 ^a,b^	68.4 ± 6.0 ^a,b^
Intact acrosome (%)	75.3 ± 1.9 ^b^	75.0 ± 2.4 ^b^	69.5 ± 2.3 ^a,b^	70.7 ± 2.1 ^a,b^	74.7 ± 1.8 ^b^	65.3 ± 3.3 ^a^	69.4 ± 3.8 ^a,b^	72.0 ± 2.4 ^a,b^
MMP (%)	58.7 ± 4.5 ^a,b^	64.5 ± 2.8 ^b^	57.0 ± 5.1 ^a,b^	61.3 ± 3.9 ^b^	64.3 ± 3.4 ^b^	43.0 ± 6.7 ^a^	51.0 ± 6.1 ^a,b^	54.3 ± 5.8 ^a,b^

Values in each row marked with different superscript letters differ significantly (*p*-value < 0.05). ABO: antibiotic (gentamicin 200 µg/mL); BTS: Beltsville Thawing Solution; MOT: total motility; PMOT: progressive motility; VCL: curvilinear velocity; VSL: velocity straight line; VAP: average pathway velocity; ALH: amplitude of lateral head displacement; BCF: beat cross frequency, straightness; STR: straightness; LIN: linearity; MMP: sperm with high mitochondrial membrane potential.

**Table 5 antibiotics-13-00489-t005:** Total bacteria count (means ± SD) from boar semen samples (n = 7) after incubated at 18 °C.

Groups	Concentrations (µg/mL)	Total Bacteria Concentration (log; CFU/mL)				
		Incubation Time				
		0 h	24 h	36 h	48 h	72 h
BTS	-	1.32 ± 0.27 ^a^	2.38 ± 0.37 ^a^	3.11 ± 0.79 ^a^	3.71 ± 0.96 ^a^	5.31 ± 1.44 ^a^
BTS + ABO	-	0.00 ± 0.00 ^b^	0.00 ± 0.00 ^b^	0.00 ± 0.00 ^b^	0.00 ± 0.00 ^b^	0.00 ± 0.00 ^b^
A-11	62.50	0.24 ± 0.16 ^b,c^	1.72 ± 0.54 ^a^	2.51 ± 0.91 ^a^	3.50 ± 1.34 ^a^	4.59 ± 1.58 ^a^
A-11	31.25	0.39 ± 0.19 ^b,c^	1.28 ± 0.53 ^a,b^	2.78 ± 0.76 ^a^	3.66 ± 1.23 ^a^	4.78 ± 1.44 ^a^
A-11	15.625	0.41 ± 0.20 ^b,c^	1.32 ± 0.63 ^a,b^	2.65 ± 0.85 ^a^	3.57 ± 1.27 ^a^	4.67 ± 1.50 ^a^
AP19	62.50	0.49 ± 0.18 ^b,c^	1.14 ± 0.54 ^a,b^	2.59 ± 0.92 ^a^	3.68 ± 1.17 ^a^	4.96 ± 1.44 ^a^
AP19	31.25	0.78 ± 0.16 ^c^	1.08 ± 0.51 ^a,b^	2.57 ± 0.70 ^a^	3.63 ± 0.99 ^a^	4.95 ± 1.22 ^a^
AP19	15.625	0.34 ± 0.17 ^b,c^	1.16 ± 0.55 ^a,b^	2.47 ± 0.90 ^a^	2.86 ± 1.35 ^a^	4.54 ± 1.44 ^a^

Values in each column marked with different superscript letters differ significantly (*p* < 0.05). ABO: antibiotic (gentamicin 200 µg/mL); BTS: Beltsville Thawing Solution.

**Table 6 antibiotics-13-00489-t006:** Physicochemical properties of the A-11 and AP19 peptides.

Peptide	Amino Acid Sequence	Number of Amino Acids	Molecular Weight (g/mol)	Net Charge	Percentage of Hydrophobic Residues
A-11	WVKKVARKVVKIGRKVAR	18	2121.66	+8	44%
AP19	RLFRRVKKVAGKIAKRIWK	19	2353.94	+9	47%

**Table 7 antibiotics-13-00489-t007:** The table shows the experimental and control groups by antimicrobial peptide type and concentration.

Group	Antimicrobial Peptides	Concentration (μg/mL)
1	BTS without gentamicin (negative control)	-
2	BTS with gentamicin 200 µg/mL (positive control)	-
3	A-11	62.50
4	A-11	31.25
5	A-11	15.625
6	AP19	62.50
7	AP19	31.25
8	AP19	15.625

## Data Availability

Data are contained within the article.

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
