# Peer review of "The Effects of Different Antimicrobial Peptides (A-11 and AP19) on Isolated Bacteria from Fresh Boar Semen and Semen Quality during Storage at 18 °C"

_antibiotics, 2024, doi:10.3390/antibiotics13060489_

Round 1

Reviewer 1 Report

Comments and Suggestions for Authors

The present study was aiming to check the inhibitory effects of two antimicrobial peptides on bacteria commonly isolated from fresh boar semen and the influence of those two antimicrobial peptides addition on boar sperm quality during liquid storage. The results indicated that the two peptides showed bactericide capability, which was comparable to gentamicin. The two peptides showed influence on sperm quality depending on storage time and their concentration. The finding provides a potential alternative for antibiotics in boar semen extender, which will be of great significance in reducing overuse of antibiotics and multidrug resistance of bacteria. Still, there are several points I would like to share my opinions with the authors.

Q1. Why did the authors choose gentamicin as the only positive control to study the effects of those two peptides on bacteria growth and sperm quality maintenance? Why not choose another peptide with proved inhibitory effect on bacteria growth?

Q2. How did the authors isolate the most common bacteria from fresh boar semen? What were the semen samples used for the survey? How many were they? How many boar studs were they from? What were the percentage of E. coli, Pseudomonas aeruginosa and Proteus mirabilis isolated from fresh boar semen? There is no related information in the manuscript.

Q3. The sperm quality parameters were measured at day 0, 1, 3 and 5 during storage, while the bacteria growth evaluation were conducted at 0, 24, 36, 48 and 72 h of storage. Why were they measured at different times?

Q4. Why did the authors choose the doses of 62.50, 31.25. 15.625μg/mL as the treatments in this study? Any data to support this design? I suggest testing the MIC and MBC of those two peptides, based on which the doses could be decided. In this study, there was only bacteria growth curve to present the inhibitory effect of bacteria. I don’t think it is enough to support the doses choice. So is gentamicin. I suggest to measure the MIC and MBC and then choose the best dose for this study. Obviously, gentamicin killed all the bacteria with 200 μg/mL. Is it the common dose in practice?

Q5. Line 312, it could be better to state clearly how many ejaculates were collected from each boar. Besides, it is not clear how many replicates were done and if pooled semen were used.

Q6. Line 347, what was the temperature when sperm motion parameters were evaluated using CASA?

Q7. Line 383, why did the authors incubate semen for 48 h to measure the bacterial growth? It is more commonly done at 12 h or 24 h in practice.

Q8. In introduction and discussion, it is suggested to do more comparison between the two peptides and other peptides from literature. And the authors should provide information on the use of these two peptides in other species or cells, and results reported in literature.

Comments on the Quality of English Language

minor

Author Response

Reviewer I

Comments and Suggestions for Authors

The present study was aiming to check the inhibitory effects of two antimicrobial peptides on bacteria commonly isolated from fresh boar semen and the influence of those two antimicrobial peptides addition on boar sperm quality during liquid storage. The results indicated that the two peptides showed bactericide capability, which was comparable to gentamicin. The two peptides showed influence on sperm quality depending on storage time and their concentration. The finding provides a potential alternative for antibiotics in boar semen extender, which will be of great significance in reducing overuse of antibiotics and multidrug resistance of bacteria. Still, there are several points I would like to share my opinions with the authors.

Q1. Why did the authors choose gentamicin as the only positive control to study the effects of those two peptides on bacteria growth and sperm quality maintenance? Why not choose another peptide with proved inhibitory effect on bacteria growth?

Responses: Thank you for your suggestions. The objective of this study is to compare the antimicrobial effects of A-11 and AP19 with the most common antibiotics used in boar semen extenders in pig industry. Therefore, gentamicin was selected as the positive control.

Q2. How did the authors isolate the most common bacteria from fresh boar semen? What were the semen samples used for the survey? How many were they? How many boar studs were they from? What were the percentage of E. coli, Pseudomonas aeruginosa and Proteus mirabilis isolated from fresh boar semen? There is no related information in the manuscript.

Responses: We appreciate reviewer’s comment. We provided more information as follows:

Lines 313-318

            The bacteria in the present study were obtained from our previous report by Keeratikunakorn et al. [7] in which three species of the most frequently found bacteria in fresh boar semen including E. coli, Pseudomonas aeruginosa, and Proteus mirabilis, were isolated and kept in a culture collection at the Laboratory of Bacteria, Veterinary Diagnostic Center, Faculty of Veterinary Science, Mahidol University (Salaya, Phuttamonthon, Nakhon Pathom, Thailand). 

Q3. The sperm quality parameters were measured at day 0, 1, 3 and 5 during storage, while the bacteria growth evaluation were conducted at 0, 24, 36, 48 and 72 h of storage. Why were they measured at different times?

Responses: Thank you for your constructive comments. The explanation are as follows:

Lines 233-241

This study used a short-term semen extender (BTS) which has ability to preserve semen quality of less than or equal to three days after dilution [34]. For the reasons mentioned, this study observed the sperm quality at days 0, 1, 3, and 5 during storage to ensure that the BTS still maintained sperm quality as claim by the manufacturer. While the bacteria growth during storage was rapid growth and significant growth after 72 h of storage, as a result, the total bacteria concentration was measured at 0, 24, 36, 48, and 72 h of storage [4, 8, 35]. In practice for the pig farms, it is also worth noted that they usually used extended boar semen within 24 h after storage. Consequently, the present experimental design was correspondent to those clinical practice.

Q4. Why did the authors choose the doses of 62.50, 31.25. 15.625μg/mL as the treatments in this study? Any data to support this design? I suggest testing the MIC and MBC of those two peptides, based on which the doses could be decided. In this study, there was only bacteria growth curve to present the inhibitory effect of bacteria. I don’t think it is enough to support the doses choice. So is gentamicin. I suggest to measure the MIC and MBC and then choose the best dose for this study. Obviously, gentamicin killed all the bacteria with 200 μg/mL. Is it the common dose in practice?

Responses: We appreciate your critical comments. The requested adjustments have been applied.

Lines 206-212

            The concentration of AMPs (A-11 and AP19) for this study came from the MIC value (62.50–15.625 µg/mL) for inhibiting Gram-negative bacteria in the previous studies [27, 28] and was further approved for inhibiting the most Gram-negative bacteria observed in fresh semen by comparing with 200 µg/mL of gentamicin [7], which is the common antibiotic mixed in boar semen extenders [13]. The results of the bacterial survival assay clearly showed the inhibitory effect of AMPs on bacterial growth in each stage of bacterial growth curve.

Q5. Line 312, it could be better to state clearly how many ejaculates were collected from each boar. Besides, it is not clear how many replicates were done and if pooled semen were used.

Responses: We much appreciate your comment. We've modified the sentences as reviewer’s suggestion.

Lines 338-339

            A semen sample was collected from each of the seven mature Duroc boars, theirs ages ranged from 1.5 to 3 years.

Q6. Line 347, what was the temperature when sperm motion parameters were evaluated using CASA?

Responses: We much appreciate your comment. We've modified incidents as suggested.

Line 363, the following sentence was added.

… in which the temperature of glass slide and stage were set at 37°C.

Q7. Line 383, why did the authors incubate semen for 48 h to measure the bacterial growth? It is more commonly done at 12 h or 24 h in practice.

Responses: Thank you for your suggestions. This method used in the present study was followed the protocol by Zead et al. (2017), and we already added the reference into manuscript.

Lines 404-405

After incubation, the colonies were enumerated and converted to CFU/mL at 48 h [48].

Q8. In introduction and discussion, it is suggested to do more comparison between the two peptides and other peptides from literature. And the authors should provide information on the use of these two peptides in other species or cells, and results reported in literature.

Responses: We appreciated reviewer’s suggestions. The recommendations were followed.

Lines 87-90

A-11 and AP19 are two novel AMPs, when used in high concentrations, are not damaging animal cells and inhibiting the growth of both Gram-positive and Gram-negative bacteria, including Salmonella enterica serovar Typhimurium and Acinetobacter baumannii [27, 28]. However, the application of these two peptides on the inhibitionof bacteria isolated from boar semen has not been reported.

Reviewer 2 Report

Comments and Suggestions for Authors

The authors of the research demonstrate the ability of synthetic compounds to inhibit bacterial growth, as well as maintain the viability of boar semen during storage. The research compares synthetic compounds with conventional antimicrobials and finds satisfactory results. The manuscript is well written in all sections (Abstract, Introduction, Methods, Results and Discussion) and can be accepted in its present form.

Author Response

Reviewer II

Comments and Suggestions for Authors

The authors of the research demonstrate the ability of synthetic compounds to inhibit bacterial growth, as well as maintain the viability of boar semen during storage. The research compares synthetic compounds with conventional antimicrobials and finds satisfactory results. The manuscript is well written in all sections (Abstract, Introduction, Methods, Results and Discussion) and can be accepted in its present form.

Responses: We appreciate your feedback and suggestions regarding this article.

Reviewer 3 Report

Comments and Suggestions for Authors

Dear Authors,

Thank you for the interesting research paper.

I believe that the explanation for why only A-11 at a concentration of 31.25 µg/mL resulted in semen quality similar to that of the positive control group should be more detailed. It would be helpful to have an illustration and a clearer description of the mechanism involved.

Author Response

Reviewer III

Comments and Suggestions for Authors

Dear Authors,

Thank you for the interesting research paper.

I believe that the explanation for why only A-11 at a concentration of 31.25 µg/mL resulted in semen quality similar to that of the positive control group should be more detailed. It would be helpful to have an illustration and a clearer description of the mechanism involved.

Responses: We appreciate your constructive suggestions. The reviewer’s recommendations are considered as follows:

Lines 249-255

The present results of A-11 clearly showed that there is no sign of toxicity to sperm cells for all concentrations. This is in agreement with the hemolytic activity examination of A-11, which discovered that A-11 did not cause damage to red blood cells at concentrations between 0.98 to 250 µg/mL [27]. The reason might be that in the outer membrane of animal cell (i.e., sperm cell) constituent of neutral components, subsequently the positively charge of AMP were not interaction with this cell [14, 22].

Lines 261-264

The negative charge of the animal cell membrane is situated internally and in close proximity to the cytoplasm and in the outer membrane of were expressed neutral components. Consequently, the positively charge antimicrobial peptides do not interact with this cell [14, 22, 27].

Reviewer 4 Report

Comments and Suggestions for Authors

Dear authors of manuscript "The Effects of Different Antimicrobial Peptides (A-11 and 2 AP19) on Isolated Bacteria from Fresh Boar Semen and Semen 3 Quality During Storage at 18°C".

I suggest that the results data provided in the tables can be presented in graphs (images) for a more agile understanding.

Your manuscript meets the scientific standards for publication after a brief review of the discussion and conclusions 

Author Response

Reviewer IV

Comments and Suggestions for Authors

Dear authors of manuscript "The Effects of Different Antimicrobial Peptides (A-11 and 2 AP19) on Isolated Bacteria from Fresh Boar Semen and Semen 3 Quality During Storage at 18°C".

I suggest that the results data provided in the tables can be presented in graphs (images) for a more agile understanding.

Your manuscript meets the scientific standards for publication after a brief review of the discussion and conclusions 

Responses: Thank you very much. We appreciate your constructive comments. However, the quality of the boar semen and the total bacteria concentration are presented in Table form because we intended to show a more details of those data for the reader to be able to compare these data for the matrix perspective.
